# Protective Effect of Daidzein against Diethylnitrosamine/Carbon Tetrachloride-Induced Hepatocellular Carcinoma in Male Rats

**DOI:** 10.3390/biology12091184

**Published:** 2023-08-29

**Authors:** Samir A. E. Bashandy, Hossam Ebaid, Jameel Al-Tamimi, Iftekhar Hassan, Enayat A. Omara, Marawan A. Elbaset, Ibrahim M. Alhazza, Jamal A. Siddique

**Affiliations:** 1Pharmacology Department, National Research Centre, 33 El-Bohouth St., Dokki, Cairo 12622, Egypt; bashandys@hotmail.com (S.A.E.B.); ma.sayed@nrc.sci.eg (M.A.E.); 2Zoology Department, College of Science, King Saud University, Riyadh 11451, Saudi Arabia; habdrabou@ksu.edu.sa (H.E.); ihazza@ksu.edu.sa (I.M.A.); 3Pathology Department, National Research Centre, 33 El-Bohouth St., Dokki, Cairo 12622, Egypt; eomara67@hotmail.com; 4Department of Materials Engineering and Chemistry, Faculty of Civil Engineering, Czech Technical University (CVUT), Praha 6, 16629 Prague, Czech Republic; sidjamal@gmail.com

**Keywords:** hepatocellular carcinoma, Daidzein, CCL4, inflammatory markers

## Abstract

**Simple Summary:**

Hepatocellular carcinoma (HCC) claims the second-largest number of casualties among all forms of cancer. Several chemotherapeutic agents are being used for its treatment, but most have been discontinued because of their side effects or the development of resistance in the patients. Hence, exploring nutraceuticals is a way to manage this disease, citing greater efficacy and a lower degree of resistance development. Daidzein (DZ), a prominent isoflavone polyphenolic phytochemical found in leguminous plants, has tremendous pharmacological properties, including anti-inflammatory, antihemolytic, and antioxidant effects. The present investigation aimed to evaluate the protective effect of DZ in DEN/CCl4-induced HCC in a rat model. The dosing of DZ was initiated four weeks before HCC induction and continued until the end of the treatment period. In this study, four treatment groups of rats (*n* = 6) were designated as control (group 1, without any treatment), HCC-induced rats (group 2), an HCC group treated with DZ at 20 mg/kg (group 3), and an HCC group treated with DZ at 40 mg/kg (group 4). Group 2 rats showed marked elevation in all the HCC markers (AFP, GPC3, and VEGF), liver function markers (ALP, ALT, and AST), inflammatory markers (IL-6, TNF-α, and CRP), and lipid markers concomitant with a decrease in antioxidant enzymes and protein. Interestingly, groups III and IV demonstrated alleviation in most of the parameters of HCC in a dose-dependent way. Also, the histological alterations of HCC were significantly reduced in groups III and IV, confirming the results of biochemical analysis. Hence, DZ is a promising candidate for HCC treatment, attributed to its antioxidant and anti-inflammatory properties.

**Abstract:**

Hepatocellular carcinoma (HCC) is the second-largest cause of death among all cancer types. Many drugs have been used to treat the disease for a long time but have been mostly discontinued because of their side effects or the development of resistance in the patients with HCC. The administration of DZ orally is a great focus to address the clinical crisis. Daidzein (DZ) is a prominent isoflavone polyphenolic chemical found in soybeans and other leguminous plants. It has various pharmacological effects, including anti-inflammatory, antihemolytic, and antioxidant. This present study investigates the protective effect of DZ on chemically induced HCC in rat models. The DZ was administered orally four weeks before HCC induction and continued during treatment. Our study included four treatment groups: control (group 1, without any treatment), HCC-induced rats (group II), an HCC group treated with DZ at 20 mg/kg (group III), and an HCC group treated with DZ at 40 mg/kg (group IV). HCC rats showed elevation in all the HCC markers (AFP, GPC3, and VEGF), liver function markers (ALP, ALT, and AST), inflammatory markers (IL-6, TNF-α, and CRP), and lipid markers concomitant with a decrease in antioxidant enzymes and protein. However, groups III and IV demonstrated dose-dependent alleviation in the previous parameters resulting from HCC. In addition, the high dose of DZ reduces many hepatological changes in HCC rats. All study parameters improved with DZ administration. Due to its antioxidant and anti-inflammatory characteristics, DZ is a promising HCC treatment option for clinical use.

## 1. Introduction

Hepatocellular carcinoma (HCC) is the most prevalent form of primary liver cancer and the leading cause of cancer-related mortality globally [1]. HCC is more prevalent among males than females, and its frequency is highest in Middle and Western Africa and Eastern and Southern Asia [2]. Males are more likely than females to be infected with viral hepatitis, smoke cigarettes, drink alcohol, and have a higher body mass index [3]. High testosterone levels are associated with HCC in hepatitis B carriers and progressive hepatic fibrosis in men with chronic hepatitis C infection [4]. Chronic liver disease and cirrhosis remain the most prominent risk factors for HCC development [5,6]. It is estimated that 74 to 80% of all liver cancers are caused by cirrhosis, which occurs from repeated viral infections caused by hepatitis B and C [7]. Alcohol is a significant risk factor in developing HCC [8]. Diabetes mellitus (DM) and obesity are two chronic medical disorders that increase the risk of HCC. DM directly affects the liver because of its critical function in glucose metabolism. It may cause chronic hepatitis, fatty liver disease, liver failure, and cirrhosis [9,10]. Aspergillus species produce aflatoxin, a potent hepatocarcinogen, during the storage of grains, maize, peanuts, or soybeans in humid and warm environments [11]. Hereditary hemochromatosis is associated with an increased risk, believed to range between 100 and 200 times, of HCC [12]. All these factors contribute to the development of HCC.

Alpha-fetoprotein (AFP) is a glycoprotein typically produced by the fetal liver and yolk sac during fetal life, while in adults, high AFP can indicate ongoing hepatic carcinoma [13]. Glypican-3 (GPC-3) is one of these promising HCC biomarkers. GPC-3 is an oncofetal protein encoded on the X chromosome [14]. GPC-3 is a member of the glypican family, a group of heparan sulfate proteoglycans joined to the cell surface through a glycosyl-phosphatidyl inositol-anchor. It has been found that glypicans interact with growth factors, modify their activities, and perform an important role in cell growth, differentiation, and migration [15,16]. GPC-3 is expressed abundantly in the fetal liver and minimally in the normal adult liver. Circulating VEGF levels are increased in HCC and have been shown to correlate with tumor angiogenesis and progression [17]. Most currently approved treatments for advanced HCC in the first- and second-line settings target angiogenic pathways. Of the known or potential angiogenic pathways in tumors, the VEGF/VEGF receptor (VEGFR) signaling pathway has been validated as a drug target in HCC [18].

Nutraceuticals comprise selected plant-derived bioactive components with medicinal, disease-preventing, and health-enhancing qualities [19]. Daidzein (DZ) is an isoflavone polyphenolic natural substance. Soybeans and other leguminous plants are primary sources of active Daidzein [20]. It is a naturally occurring phytoestrogen classified as nonsteroidal estrogen [21]. Daidzein has various pharmacological effects, including anti-inflammatory, antihemolytic, and antioxidant effects [22]. It possesses potent free-radical scavenging and antioxidant properties [23]. Miyake et al. [24] suggested that eating soy and taking Daidzein may help Japanese ladies with allergic rhinitis. In addition, Daidzein has been shown to have both anti-inflammatory and neuroprotective effects against oxidative-stress-induced Parkinson’s disease in animal models [25]. Furthermore, Daidzein is used in treating cerebral ischemia because of its neuroprotective effects against oxygen–glucose deficiency-induced neurotoxicity and glutamate-induced excitotoxicity in brain cells [26,27]. In a rat model, the protective effect of Daidzein against streptozotocin-induced Alzheimer’s disease was demonstrated by an improvement in cognitive impairment and oxidative stress [28]. Shah et al., [29] suggested that Daidzein plays an essential role in treating ovarian ischemia. Daidzein has been shown to have anticancer effects in many distinct types of cancers, according to several in vitro studies [30,31,32,33]. Following an extensive examination of the existing literature, we were unable to find any instances where Daidzein was reported to possess protective capabilities against HCC. The objective of the present research was to appraise the potential protective properties of Daidzein against HCC resulting from diethylnitrosamine/carbon tetrachloride exposure. This evaluation encompassed the analysis of oxidative stress indicators, inflammatory markers, and HCC-related indicators.

## 2. Materials and Methods

### 2.1. Material

Daidzein was purchased from Shaanxi Sciphar Biotechnology Co., Ltd., Xi’an, China. Tween 80, Diethylnitrosamine (DENA), and carbonate tetrachloride (CCl4) were purchased from Sigma Aldrich (St. Louis, MO, USA).

### 2.2. Animals and Treatment

Adult male Wistar rats (weighing 150–170 g and 3 months’ age) were bought from the National Research Centre (NRC), Egypt, Cairo. They were fed a standard pellet diet and water ad libitum and kept at an adjusted temperature (22 ± 2 °C) with a 12 h light–dark cycle. Animal handling was carried out following the recommendations of the National Institute of Health Guide for Care and Use of Laboratory Animals (Publication No. 85–23, revised 1985) and approved by institutional Review committee of the NRC (Reg. No. (13114052023). All sacrifice was performed under anesthesia, and all efforts were made to reduce suffering.

Twenty-four rats were divided into four groups:

Group 1: Normal rats served as control and received equivalent volume of vehicle (water in 0.1% Tween 80). 

Group 2: Rats received DENA/CCl4 to induce HCC and administered equivalent volume of vehicle (water in 0.1% Tween 80).

Group 3: Rats received DENA/CCl4 and were treated orally with 20 mg/kg Daidzein in (0.1% Tween 80) four weeks prior to the DENA and for the next 8 weeks [34].

Group 4: Rats received DENA/CCl4 and were administered 40 mg/kg Daidzein orally in (0.1% Tween 80) four weeks prior to the DENA injection and for the next 8 weeks.

The administration of Daidzein orally was initiated four weeks before DENA injection and continued daily for the next 8 weeks of its injection.

### 2.3. Induction of Hepatocellular Carcinoma (HCC)

Diethylnitrosamine (DENA) was dissolved in physiologic saline solution (0.9% NaCl) and injected intraperitoneally into each rat in a single dose of 200 mg/kg body weight. After two weeks, animals received CCl4 (3 mL/kg) injected subcutaneously once a week for 6 weeks to develop hepatocellular carcinoma [35]. 

### 2.4. Samples Collection

The blood was collected at the end of the experiment in heparinized tubes from the tail vein, and plasma was isolated by centrifugation at 3000× *g* for 15 min. The animals were killed after anesthesia to collect tissue samples, and a portion of the liver samples were processed for histological study. The samples were further subjected to the following analyses.

### 2.5. Liver Function Tests

Plasma AST, ALT.GGT, LDH, and ALP were determined colorimetrically using kits manufactured by Spectrum, the Egyptian company for Biotechnology, Cairo, Egypt.

### 2.6. Hepatic Oxidative Stress Parameters

Malondialdehyde (MDA), reduced glutathione (GSH), catalase (CAT), and nitric oxide (NO) were assayed colorimetrically in liver tissue using Biodiagnostic kits, Giza, Egypt.

### 2.7. Lipid Profile

Plasma total cholesterol (TC), triglycerides (TG), LDL, and HDL were estimated using kits manufactured by Spectrum, the Egyptian company for Biotechnology.

### 2.8. Inflammatory Markers

Plasma CRP, IL6, and TNF-a were determined with the ELISA technique using kits from Elabscience Com., China.

### 2.9. Plasma Cancer Markers

Plasma alpha-fetoprotein (AFP) was determined by the ELISA technique using Sunlong Biotech Co., Ltd., China kits. At the same time, VEGF and glypican-3 (GPC3) were assayed by ELISA technique using kits purchased from Lifespan Biosciences, Inc., Seattle, WA, USA.

### 2.10. Histopathological Studies

The liver samples were fixed in 10% neutral buffered formalin dehydrated with 100% ethanol solution and embedded in paraffin. They were then processed into 5 μm thick sections stained with hematoxylin–eosin and observed under a photomicroscope.

### 2.11. Statistical Analysis

Data were evaluated by one-way ANOVA followed by Bonferroni multiple comparisons. The level of significance was accepted at *p* < 0.05. The degree of variability of results was expressed as means ± standard error of means (SEM). GraphPad Prism (v5) was used to draw the graphs.

## 3. Results

### 3.1. Effect of Daidzein on Body Weight in Rats Dosed with DENA/CCl4 against Diethylnitrosamine/Carbon Tetrachloride-Induced Hepatocellular Carcinoma in Male Rats

Data showed that DENA/CCl4 significantly lowered body weight compared to normal rats (*p* < 0.05). A significant improvement was observed in the body weight of DENA rats treated with Daidzein in low or high doses (*p* < 0.05) compared to the DENA/CCl4 group. On the contrary, the relative liver weight in DENA/CCl4 rats was significantly higher than the normal ones (*p* < 0.05), indicating an increase in the weight of liver tissues due to HCC induction. An improvement in the liver-to-body-weight ratio of DENA/CCl4 rats (Daidzein treated) was observed in low or high doses compared to the control rats (*p* < 0.05). This result indicates that Daidzein inhibited the neogenesis of the HCC in the liver tissues of DENA rats (Figure 1).

### 3.2. Effect of Daidzein on Plasma Liver Function Tests in Rats Intoxicated with DENA/CCl4

Estimating the liver functions from animals treated with DENA/CCl4 showed a significant increase in ALP, ALT, and AST compared to the control rats (*p* < 0.05). Daidzein significantly restored all the liver functions (ALP, ALT, and AST) in DENA/CCl4 rats in a dose-dependent manner (*p* < 0.05). Interestingly, both groups of Daidzein (20 or 40 mg/kg) showed substantially lower levels of ALP, ALT, and AST (*p* < 0.05) than the HCC group (Figure 2). 

### 3.3. Effect of Daidzein on Liver Content of Oxidative Stress and Antioxidative Parameters in Rats Intoxicated with DENA/CCl4

Malondialdehyde (MDA) and nitric oxide (NO) levels in the hepatic tissue of the HCC group were significantly higher (*p* < 0.05) than those in the control group. Daidzein 20 or 40 mg/kg substantially hampered (*p* < 0.05) the MDA and NO hepatic content in a dose-dependent manner relative to the HCC group (Figure 3).

### 3.4. Effect of Daidzein on Liver Content of Reduced Glutathione (GSH) and Catalase (CAT) in Rats Intoxicated with DENA/CCl4

GSH (glutathione) is a crucial intracellular antioxidant and serves as a primary cellular reductant to assess oxidative stress levels in vivo. Here, DENA/CCl4 rats showed a significant decrease (*p* < 0.05) in both the level of GSH and CAT activity of the control group. Daidzein-treated group III and group IV exhibited a significant restoration (*p* < 0.05) in GSH hepatic content in a dose-dependent manner compared to the HCC group. The high dose of Daidzein restored the hepatic content of the reduced glutathione to a level close to the normal value; however, it was not statistically significant.

Catalase is an important antioxidant enzyme that plays a vital role in the defense against oxidative stress in living organisms. In the present work, group II, DENA/CCl4, showed a significant reduction (*p* < 0.05) in GSH hepatic content and hepatic CAT activity with respect to the control group. Daidzein-treated groups III and IV significantly restored the GSH hepatic content and the hepatic CAT activity (Figure 4) compared to the HCC group.

### 3.5. Effect of Daidzein on Lipid Profile in Rats Intoxicated with DENA/CCl4

All the lipid parameters (TC, TG, and LDL) showed a significant reduction (*p* < 0.05) in the DZ-treated groups in a dose-dependent manner compared to the HCC group. However, DZ also elevated (*p* < 0.05) HDL levels in the treated groups dose-dependently compared to the HCC group (Figure 5). 

### 3.6. Effect of Daidzein on Inflammatory Factors in Rats Intoxicated with DENA/CCl4

Estimating the inflammatory factors from animals treated with DENA/CCl4 showed a significant increase (*p* < 0.05) in IL-6, TNF-α, and C-reactive protein (CRP) compared to those in the control rats. Daidzein was found to significantly (*p* < 0.05) mitigate all inflammatory markers in DENA/CCl4 rats treated with both low and high doses of Daidzein in a dose-dependent manner relative to the control rats (Figure 6).

### 3.7. Effect of Daidzein on Gamma-Glutamyl Transferase (GGT) in Rats Intoxicated with DENA/CCl4

GGT is a prominent marker for assessing target organ toxicity [35]. In the present study, the DENA/CCl4 group demonstrated an increase (*p* < 0.05) in activity compared to control animals. However, Daidzein showed a significant (*p* < 0.05) decrease in its activity compared to the DENA/CCl4 group (Figure 7).

### 3.8. Effect of Daidzein on the Activity of Lactate Dehydrogenase in Rats Intoxicated with DENA/CCl4

Lactate dehydrogenase (LDH) indicates necrosis in the living system [36]. In the present study, group II, DENA/CCl4, showed a significant (*p* < 0.05) increase in LDH activity compared to control. The LDH activity in rats of groups III and IV displayed a significant (*p* ≤ 0.05) reduction in activity, respectively, as compared with the DENA/CCl4 group (Figure 8). Nonetheless, a high dosage of Daidzein considerably (*p* < 0.05) reduced LDH activity compared to the HCC group and to a level equivalent to that of the control group.

### 3.9. Effect of Daidzein on HCC Markers in Rats Intoxicated with DENA/CCl4

Serum GPC-3 has a potential sensitivity for diagnosing HCC [15]. Estimation of the HCC markers in animals treated with DENA/CCl4 exhibited significant (*p* < 0.05) elevation in the AFP, GPC-3, and VEGF compared to their concentrations in the control rats. The treatment with Daidzein was found to significantly alleviate (*p* < 0.05) these markers in DENA/CCl4 rats (Figure 9) relative to the HCC group.

### 3.10. Histopathology of the Hepatic Tissues

Sections in control liver tissue demonstrate normal histology; the central vein and hepatocyte cords radiated from the central vein and were separated by hepatic blood sinusoids and central rounded vesicular nuclei (Figure 10A). The present study was aimed at investigating the protective effect of DZ by evaluating histopathological changes. The examination of liver sections from animals treated with DENA/CCl4 showed alterations as loss of normal architecture of hepatic lobules, ballooning degeneration of hepatocytes, remarkable microvesicular and macrovesicular steatosis, cell necrosis, dilations in the central vein, and a more significant number of collagen fibers with infiltration of inflammatory cells, with pyknotic nuclei as compared to the control group (Figure 10B,C). Liver sections of low dose of Daidzein showed a mild ameliorative effect; however, hepatocytes in most fields showed a decrease in microvesicular and macrovesicular steatosis. Central vein and blood sinusoids became less dilated and less congested with the restoration of hepatic cords. However, some areas still had several collagen fibers, with infiltration of inflammatory cells with pyknotic nuclei seen surrounding the dilated central vein (Figure 10D). The high dose of Daidzein showed a remarkable ameliorative effect, as shown in liver sections from DENA/CCl4 rats with more or less normal hepatic architecture in many fields. The central vein, still congested, was surrounded by branched cords of liver cells and surrounded with mild inflammatory cells; few hepatocytes still showed small pyknotic nuclei. Hepatocytes in most fields showed a marked decrease in microvesicular and macrovesicular steatosis (Figure 10E).

## 4. Discussion

Nowadays, nutraceuticals are of great importance for their plant-derived bioactive components with medicinal properties, disease-preventing capabilities, and health-enhancing properties. Regarding phytochemical-rich plant sources, especially concerning isoflavonoid sources, soybeans and other leguminous plants are the primary sources of active isoflavones, including genistein and Daidzein [20,37]. Daidzein (DZ) is a natural ingredient considered as phytoestrogen under the category of nonsteroidal estrogens [38]. It contains extensive pharmacologically relevant properties, including antihemolytic, antioxidant, and anti-inflammatory activities [39,40]. The present study aimed to investigate the protective efficacy of Daidzein against chemical-induced HCC in a rat animal model.

In the present study, all the liver function tests (ALP, ALT, and AST), GGT, LDH, lipid markers (HDL, HDL, cholesterol, TAGs), macromolecular oxidation products (MDA, NO), and inflammatory markers (IL-6, TNF-α, and CRP) were prominently elevated in the HCC rat group as compared to the normal, while the cellular redox markers (CAT, GSH) were highly compromised in the same. Also, tumor markers (AFP, GPC3, and VEGF) were significantly enhanced in the HCC group. Interestingly, DZ improved all the parameters towards the control in a dose-dependent way. The histology of liver samples also showed agreement with the serum and tissue samples analysis for all the groups. DZ treatment improved liver function and reduced hepatic pathological changes in HCC rats by alleviating the increase of inflammatory markers. Chronic inflammation is considered a significant risk factor for cancer formation. Inflammation within the tumor environment affects its response to therapy, growth, and prognosis. TNF-α is a crucial inflammatory cytokine in the development of liver disease. This cytokine can cause hepatic injury, cirrhosis, and eventually promote hepatocellular carcinoma [41]. Also, CRP and IL2 are associated with HCC recurrence [42,43].

It is well established that DZ has tremendous antioxidant and anti-inflammatory properties [44] that help restore all the studied parameters toward the control values. The carcinogenesis triggered by DEN/CCl4 involves liver cells, particularly Kupffer cells. As carcinogens generate free radicals (ROS and RNS), they elevate the oxidative stress in the target cells, including the liver and Kupffer cells. These radicals further invade or react with macromolecules (lipids and proteins) that disrupt the cell organelles, including the nucleus; hence, DNA and RNA are damaged. The accumulation of the radicals further triggers inflammation and the combined effects of all these events, consequently leading to the causation of HCC. Alternatively, stellate cells are also transformed into myofibroblasts that cause fibrosis leading to HCC. On the contrary, DZ enhances antioxidant status, elevating the activity of GSH and catalase concurrent with the release of anti-inflammatory cytokines. These events cease the derogatory impacts of free radicals and control inflammation. Hence, DZ reverses all the events triggered by the carcinogen in the target cell in the present study.

Numerous studies indicate that DZ induces considerable anticancer action via the development of programmed cell death [45]. The compound has been reported to trigger apoptosis via the mitochondrial apoptotic pathway in various cancer types, such as breast cancer, gastric carcinoma, and hepatic cancer, by optimizing the Bcl-2/Bax ratio to activate the caspase cascade [46]. Also, multiple derivatives of the compound have been shown to promote apoptosis in colon adenocarcinoma and hepatocellular carcinoma cells [47]. Further, DZ has been reported to be a potent inhibitor of cyclins and cyclin-dependent kinases (CDKs) regulation that arrests the cell cycle progression at the G0/G1 phase [48]. The family of such proteins negatively regulates cyclin/CDK complexes that help to maintain a balance between cell proliferation and apoptosis in normal cells [49]. Chen et al. (2000) [50] reported that these proteins induced upregulation of two CDKIs, p27, and p21, in prostate cancer cell lines. Also, it is documented that DZ promotes the induction of apoptosis via various pathways, including the extrinsic receptor-mediated pathway, intrinsic mitochondrial pathway, or endoplasmic reticulum stress pathway; however, the pathway of cell death operates depending on the type of cancer [32]. The various forms of this compound have been shown to actuate different pathways of programmed cell death depending on cancer cell type [42]. In most studies, DZ has been found to involve Caspase-9, and the Bcl-2/Bax ratio has been projected as a major activator of the mitochondrial apoptotic pathway [51]. Additionally, recent studies indicate that the compound can enhance the anticancer efficacy of established drugs against various cell lines via increasing chemosensitivity [52]. The putative mechanism of the chemoprotective effect of Daidzein against chemical-induced HCC in vivo is shown in Figure 11.

Furthermore, Ref. [53] reported that the compound triggers the upregulation of Bax and downregulation of Bcl2. In addition, an intense burst of ROS triggered by DZ indicates the induction of the intrinsic pathway of apoptosis. Hence, DZ shows anticancer activity inhibiting cell cycle, cell growth, angiogenesis, metastasis, apoptotic process, and epigenetic modifications mediated through different signaling pathways [32].

## 5. Conclusions

Hepatocellular carcinoma (HCC) stands as a significant contributor to liver disease-related mortality, often hindered by discontinuation of conventional treatments due to adverse effects and resistance development. This has prompted a shift towards exploring nutraceutical interventions to address this clinical dilemma. Daidzein (DZ), a potent isoflavone phenolic compound abundant in soybeans and leguminous plants, holds diverse pharmacological properties including anti-inflammatory, antihemolytic, and antioxidant effects.

This study delved into DZ’s potential as a protective agent against chemically induced HCC in rat models. Through oral administration initiated four weeks prior to HCC induction and sustained during treatment, the research encompassed four distinct treatment groups. HCC-induced rats exhibited elevated HCC markers, liver function indicators, inflammatory markers, and lipid markers, accompanied by a decline in antioxidant enzymes and proteins. However, groups treated with DZ at 20 mg/kg (group III) and 40 mg/kg (group IV) displayed dose-dependent mitigation across these parameters, effectively countering the HCC-induced changes. Particularly, the higher DZ dosage yielded significant improvements in various hepatological alterations within HCC rats.

The collective improvement across all study parameters with DZ administration underscores its potential as a promising therapeutic option for HCC treatment. By virtue of its antioxidant and anti-inflammatory properties, DZ offers a compelling avenue for future clinical utilization in addressing hepatocellular carcinoma.

## Figures and Tables

**Figure 1 biology-12-01184-f001:**
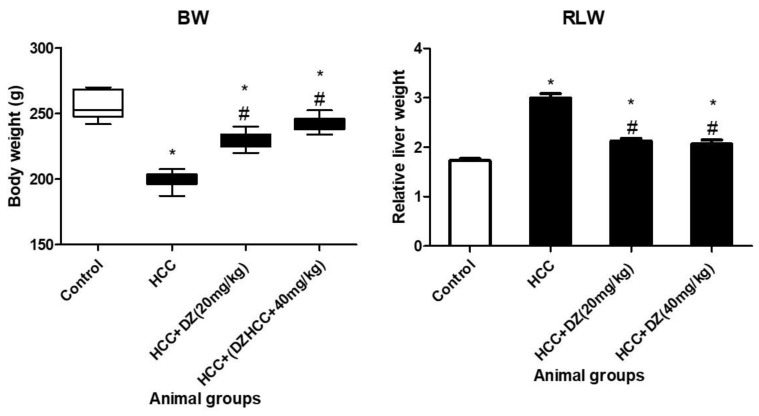
Average body weight (BW) and relative loss of weight (RLW) of the treatment groups in grams. * Indicates statistically different from the control at *p* ≤ 0.5, while # indicates statistically different from the HCC group.

**Figure 2 biology-12-01184-f002:**
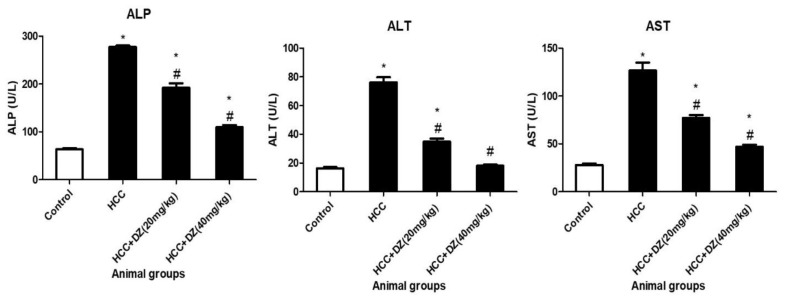
Effects of Daidzein on liver function markers (ALP, ALT, and AST) in rats intoxicated with DENA/CCl4 are shown as mean ± SD (*n* = 5–6) in units/liter. * Indicates statistically different from the control at *p* ≤ 0.5, while # indicates statistically different from the HCC group.

**Figure 3 biology-12-01184-f003:**
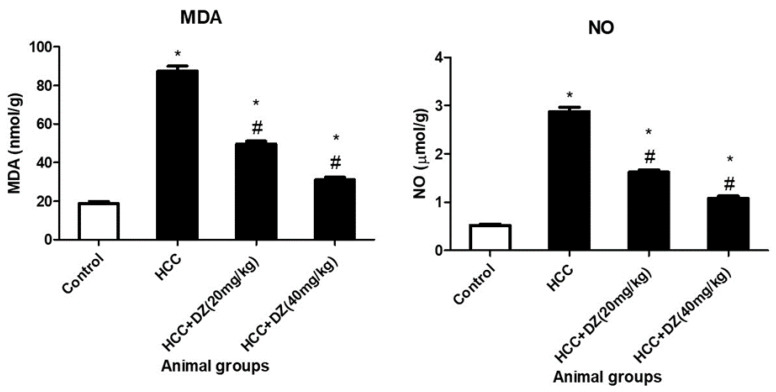
Effects of Daidzein on liver oxidative stress markers (MDA and NO) in rats intoxicated with DENA/CCl4 (*n* = 5–6) are shown as mean ± SD. * Indicates statistically different from the control at *p* ≤ 0.5, while # indicates statistically different from the HCC group.

**Figure 4 biology-12-01184-f004:**
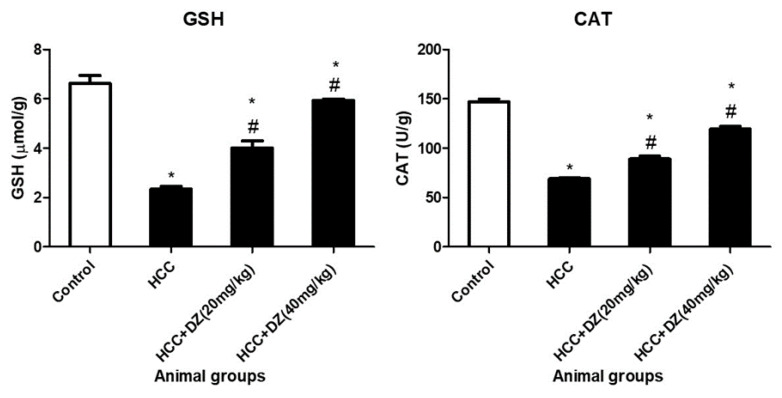
Effects of Daidzein on liver antioxidative parameters (GSH and CAT) in rats intoxicated with DENA/CCl4 (*n* = 5–6) are shown as mean ± SD. * Indicates statistically different from the control at *p* ≤ 0.5, while # indicates statistically different from the HCC group.

**Figure 5 biology-12-01184-f005:**
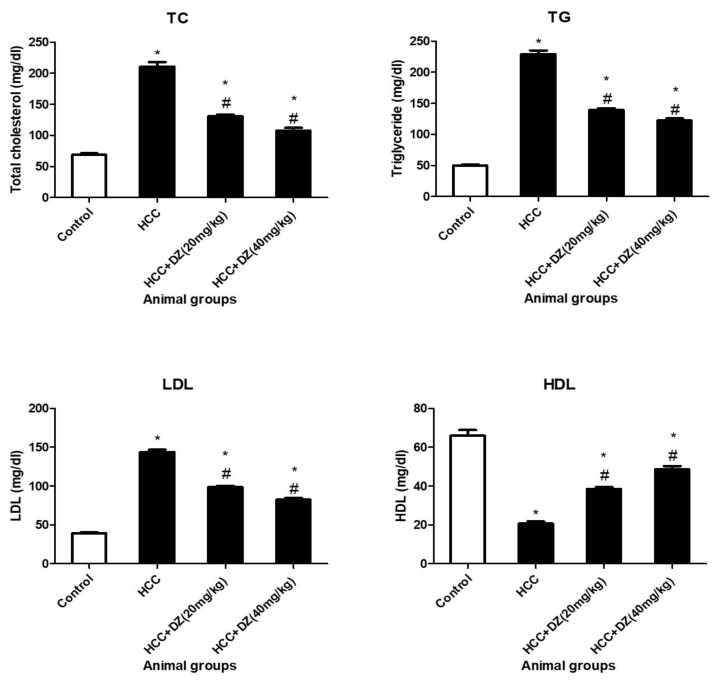
Effects of Daidzein on lipid profile parameters (TC, TG, LDL, and HDL) in rats intoxicated with DENA/CCl4 (*n* = 5–6) shown as mean ± SD. * Indicates statistically different from the control at *p* ≤ 0.5, while # indicates statistically different from the HCC group.

**Figure 6 biology-12-01184-f006:**
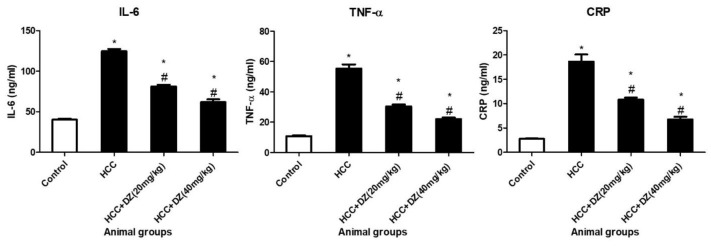
Effects of Daidzein on plasma inflammatory factors (IL-6, TNF-α, and CRP) in rats intoxicated with DENA/CCl4 (*n* = 5–6) are shown as mean ± SD. * Indicates statistically different from the control at *p* ≤ 0.5, while # indicates statistically different from the HCC group.

**Figure 7 biology-12-01184-f007:**
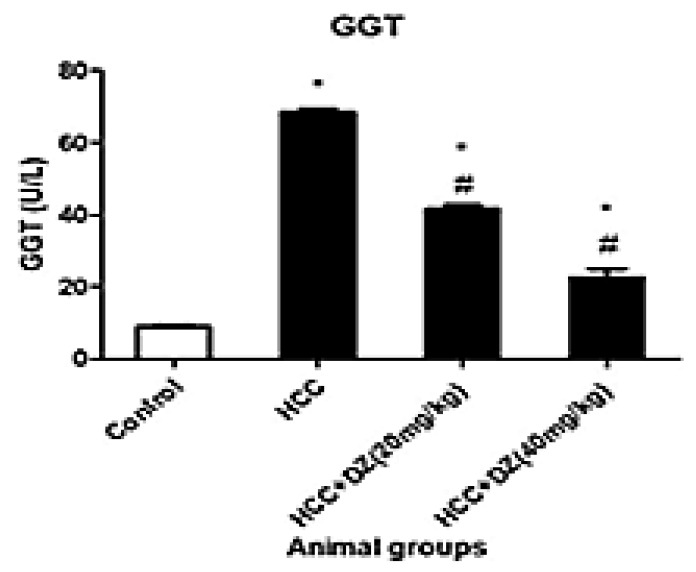
Effects of Daidzein on plasma GGT activity, LDH activity, and glucose concentration in rats intoxicated with DENA/CCl4 (*n* = 5–6) shown as mean ± SD. * Indicates statistically different from the control at *p* ≤ 0.5, while # indicates statistically different from the HCC group.

**Figure 8 biology-12-01184-f008:**
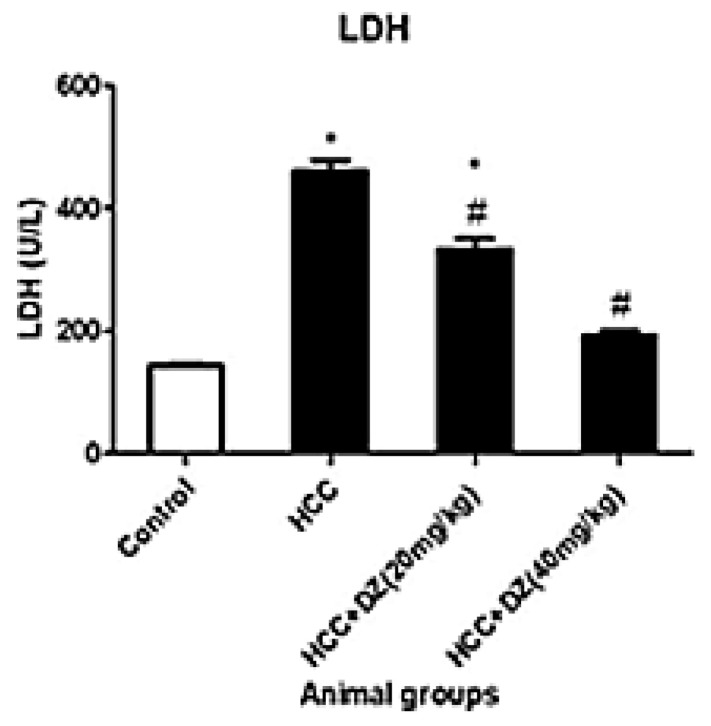
Effects of Daidzein on plasma LDH activity in rats intoxicated with DENA/CCl4 (*n* = 5–6) shown as mean ± SD. * Indicates statistically different from the control at *p* ≤ 0.5, while # indicates statistically different from the HCC group.

**Figure 9 biology-12-01184-f009:**
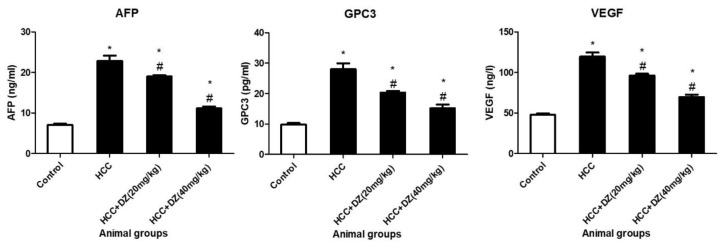
Effects of Daidzein on AFP, GPC3, and VEGF in rats intoxicated with DENA/CCl4 (*n* = 5–6) shown as mean ± SD. * Indicates statistically different from the control at *p* ≤ 0.5, while # indicates statistically different from the HCC group.

**Figure 10 biology-12-01184-f010:**
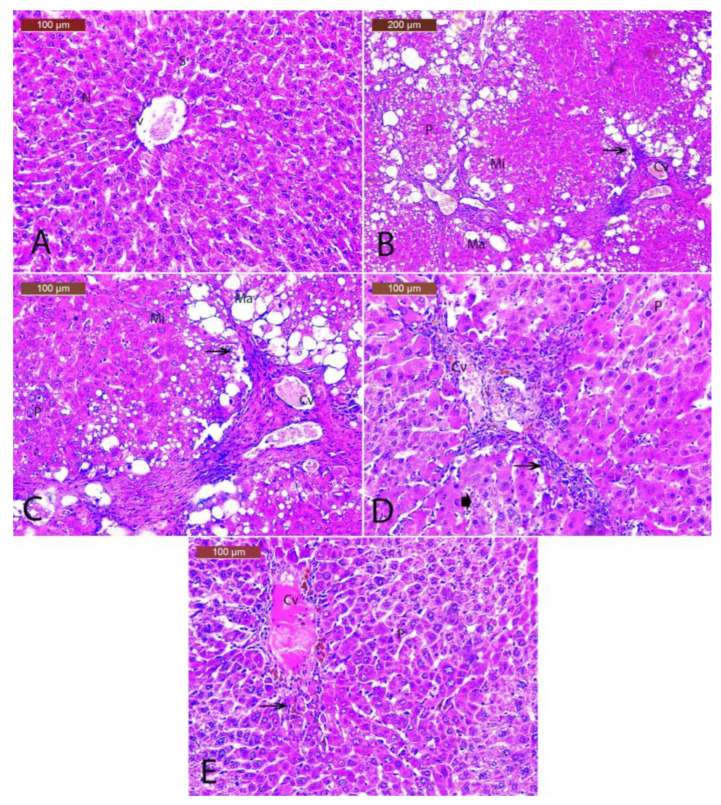
(**A**) Photomicrographs of the liver of a rat in the control group show the central vein (Cv), hepatic cords, blood sinusoids (S), and central rounded vesicular nuclei (N). (**B**) Photomicrographs of the liver from the DENA/CCl4 group show that histopathological changes were found in the form of loss of normal architecture of hepatic lobules, ballooning degeneration of hepatocytes, remarkable microvesicular (Mi) and macrovesicular (Ma) steatosis, cell necrosis, dilations and congestion in the central vein (Cv), and a larger number of collagen fibers with infiltration of inflammatory cells (P). (**C**) High magnification of DENA/CCl4 group showing histopathological changes in the form of loss of normal architecture of hepatic lobules, ballooning degeneration of hepatocytes, remarkable microvesicular (Mi) and macrovesicular (Ma) steatosis, cell necrosis, dilations congestion in the central vein (Cv), and a more significant number of collagen fibers with infiltration of inflammatory cells (arrow) with py (P). (**D**) Photomicrographs of the liver of a rat given DENA/CCl4 and a low dose of Daidzein show a mild improvement: a decrease in the microvesicular (Mi) and macrovesicular (Ma) steatosis, a less swollen and clogged central vein, and the return of the hepatic cords. However, there are still areas with collagen fibers and inflammatory cells with pyknotic nuclei (P). (**E**) Photomicrographs of the liver of a rat given DENA/CCl4 and a high dose of Daidzein show a remarkable improvement. The liver’s structure looks normal in many ways. The central vein is still congested (Cv), and there are a few inflammatory cells (arrow) and small pyknotic nuclei in a few hepatocytes (P).

**Figure 11 biology-12-01184-f011:**
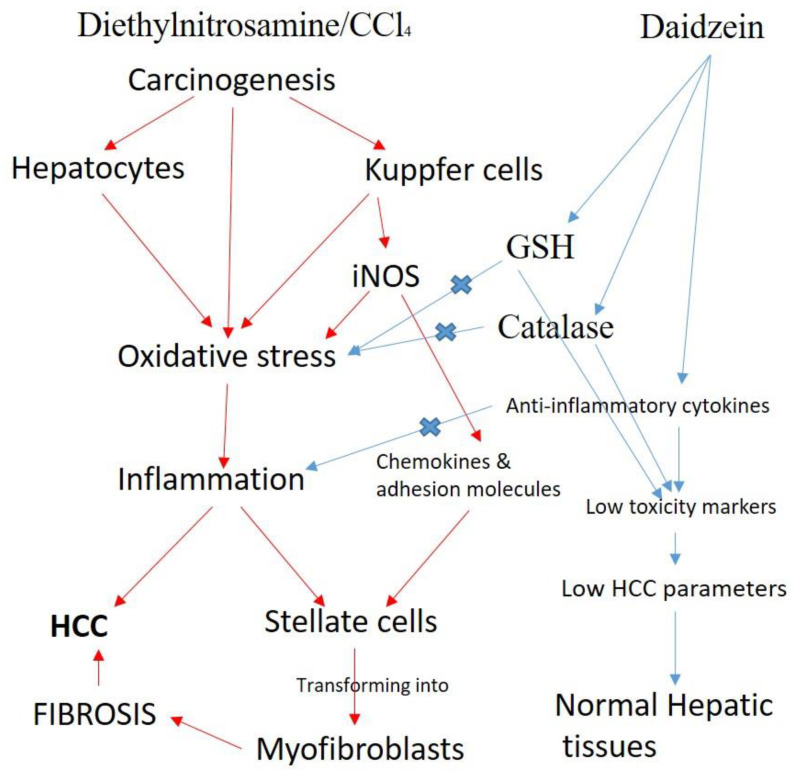
Putative mechanism of chemoprotective effect of Daidzein against chemical-induced HCC in vivo. Red arrow: disease injurious stimulation; blue arrow; protective drug action and X means inhibition of injurious action.

## Data Availability

All data generated or analyzed during this study are included in this published article.

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
