# Peer review of "Protective Effect of Daidzein against Diethylnitrosamine/Carbon Tetrachloride-Induced Hepatocellular Carcinoma in Male Rats"

_biology, 2023, doi:10.3390/biology12091184_

Round 1
Reviewer 1 Report
In the current manuscript, the authors discuss the anticancer effect of Daidzein as anti-HCC therapy in a rat model. It's an acceptable study idea, and the references are recent. The methodology, results, and discussion were poorly written. The paper plan is missing. I think the authors collected their evidence without giving them a structure. My general response is that the authors must rewrite the manuscript from the beginning. Please find below a resume of my criticism and comments:
1. In the abstract, “Daidzein (DZ) is a prominent isoflavone polyphenolic chemical found in soybeans and other leguminous plants” I think “chemical” didn’t suitable, it is better to be “Daidzein (DZ) is a prominent isoflavone polyphenolic constituent found in soybeans and other leguminous plants”.
2. Authors claimed “After treatment, HCC rats showed elevation in all the HCC markers (AFP, GPC3 and VEGF), liver function markers (ALP, ALT, AST), inflammatory markers (IL-6, TNF-α, CRP), and lipid markers concomitant with a decrease in antioxidant enzymes and protein” from this sentence I understand that treatment with DZ had elevated the HCC markers!! Please carefully revise the abstract section.
3. The introduction section:
Line number 48, “Alpha-fetoprotein (AFP) is a glycoprotein typically produced by the fetal liver and yolk sac during fetal life, while in adults, high AFP can indicate ongoing disease.” What disease is this? And high AFP may be found in inflammatory conditions, viral diseases, or cancers. The author must be specified in scientific information.
4. In line number 62 “Daidzein is an isoflavone polyphenolic chemical” It is a natural compound, not a chemical compound. Please review.
5. The aim of the study needs improvement.
2. In the methodology section,
· The animal approval number is missing, and the written “(Publication No. 85-23, revised 1985)” is about the published Principles of laboratory animal use and Care.
· “All surgery was performed under anesthesia and all efforts were made to reduce suffering” What is the type of surgery done in this study? Authors must clarify.
· Please add a reference to the metformin dose.
· “Group 2: Rats treated with DENA/CCl4”, Is DENA/CCl4 a treatment? The carcinogen must be written as administrated or injected not treated.
· “Group 3: Rats treated with DENA/CCl4 and administered orally 20 mg/Kg daidzein” Please revise.
· The (Karale and Kamath, 2017) reference is absent from the reference list.
· “The administration of Daidzein orally was initiated four weeks before HCC induction and continued daily until the end of the treatment period." The authors did not mention the study period or the treatment period.
- Daidzein's solvent is missing. And the solvent group is missing.
· The authors must rewrite the animal groping and treatment periods to be understood by the reader.
· The authors didn’t clarify the sample type (serum or liver tissue) used in the oxidative stress markers.
· Did GGT, LDH, and glucose considered toxicity markers?
3. In the results:
· The p values of the group comparisons were missing.
· In the results, the authors named a group the positive control group, and this group isn’t found in the methodology section. Also, the control group is written control and is written group I. The author must uniform the group naming to not confuse the reader.
· “GSH is a vital antioxidant and a prominent cellular reductant to assess oxidative stress in vivo”, and “Catalase is a vital antioxidant enzyme in vivo". This is not scientific writing.
· The results were poorly written.
· In lines number 228 “Lactate dehydrogenase (LDH) indicates necrosis in the living system [36].” Reference number 36 is about the detection of necrosis by the release of lactate dehydrogenase activity from cell culture media not in animal serum.
· In lines number 232 “However, the high dose of Daidzein was found to significantly restore the LDH activity to a value close to that of the control rats” What is meant by close to?
· In lines number 258 “Liver sections of DENA and low dose of Daidzein showed mild ameliorative effect” Is DENA the group I? This is very confusing and difficult to understand.
· Fig. 10 legends need simplification.
4. In the discussion section:
· The authors write Daidzein (DZ) abbreviation in the introduction section and used that abbreviation in some places in the manuscript however referred to this abbreviation three times in the results. It must be written first, then used throughout the manuscript.
· The authors did not refer to Fig. 11 in the text.
According to the authors, DZ has tremendous antioxidant and anti-inflammatory properties [41] while reference number 41 describes rat hepatocellular carcinoma induced by these chemicals. Please revise the references.
· In lines number 324-325, “These events cease the derogatory impacts of free radicals and control the inflammation These events cease the derogatory impacts of free radicals and control the inflammation. Hence, DZ reverses all the events triggered by the carcinogen in the target cell in the present study”, the word events were used twice and contrary to each other, which is very confusing.
· “A great deal of literature suggests that DZ suggests a significant level of antitumor activity” This is a vague sentence, and only one reference [42] was added!
· “[47] reported that these proteins induced up-regulation of two CDKIs, p27 and p21, in prostate cancer cell lines” and “Furthermore, [50] have reported that the compound triggers the upregulation” Please revise.
· The discussion needs to be improved.
· The Conclusions section is very poor. I only found two conclusive sentences, but what is reported is not consistent with what I understood from the results:
" In conclusion, the present investigation demonstrates that DZ causes apoptosis induction in the chemically induced HCC in rat models via the mitochondrial apoptotic pathway in a dose-dependent manner (Fig. 11).”, also, no figure in the conclusion section.
I added the language comments with other reported comments.
Author Response
Response to comments of Reviewer 1
The authors are thankful for careful evaluation of our manuscript. We have tried to incorporate the suggestions in the revised version wherever applicable. Please find the point-to point response (green text) against each comment (red text) in the below.
In the current manuscript, the authors discuss the anticancer effect of Daidzein as anti-HCC therapy in a rat model. It's an acceptable study idea, and the references are recent. The methodology, results, and discussion were poorly written. The paper plan is missing. I think the authors collected their evidence without giving them a structure. My general response is that the authors must rewrite the manuscript from the beginning. Please find below a resume of my criticism and comments:
- In the abstract, “Daidzein (DZ) is a prominent isoflavone polyphenolic chemical found in soybeans and other leguminous plants” I think “chemical” didn’t suitable, it is better to be “Daidzein (DZ) is a prominent isoflavone polyphenolic constituent found in soybeans and other leguminous plants”.
Response: The referred line has been corrected as suggested.
- Authors claimed “After treatment, HCC rats showed elevation in all the HCC markers (AFP, GPC3 and VEGF), liver function markers (ALP, ALT, AST), inflammatory markers (IL-6, TNF-α, CRP), and lipid markers concomitant with a decrease in antioxidant enzymes and protein” from this sentence I understand that treatment with DZ had elevated the HCC markers!! Please carefully revise the abstract section.
Response: The authors have revised the abstract section and adjusted it.
- The introduction section:
Line number 48, “Alpha-fetoprotein (AFP) is a glycoprotein typically produced by the fetal liver and yolk sac during fetal life, while in adults, high AFP can indicate ongoing disease.”
What disease is this? And high AFP may be found in inflammatory conditions, viral diseases, or cancers. The author must be specified in scientific information.
Response: The sentence has been elaborated for clarity to the readers.
- In line number 62 “Daidzein is an isoflavone polyphenolic chemical” It is a natural compound, not a chemical compound. Please review.
Response: Corrected
- The aim of the study needs improvement.
Response: The aim has been improved as per suggestion.
- In the methodology section,
The animal approval number is missing, and the written “(Publication No. 85-23, revised 1985)” is about the published Principles of laboratory animal use and Care.
“All surgery was performed under anesthesia and all efforts were made to reduce suffering” What is the type of surgery done in this study? Authors must clarify.
Response: The drafting authors got confused between ‘sacrifice’ with ‘surgery’. We state that no surgery was made. However, after completion of the treatment, animals were sacrificed as per the Internationally accepted ethical methods. The relevant sentence has been rephrased in the revision.
Please add a reference to the metformin dose.
Response: Metformin is not used in dosing the test compounds in the present manuscript.
- “Group 2: Rats treated with DENA/CCl4”, Is DENA/CCl4 a treatment? The carcinogen must be written as administrated or injected not treated.
Response: Corrected as per the comment.
- “Group 3: Rats treated with DENA/CCl4 and administered orally 20 mg/Kg daidzein” Please revise.
Response: Corrected as per the comment.
- The (Karale and Kamath, 2017) reference is absent from the reference list.
Response: the references were adjusted.
- “The administration of Daidzein orally was initiated four weeks before HCC induction and continued daily until the end of the treatment period." The authors did not mention the study period or the treatment period.
Response: Corrected. The administration of Daidzein orally was initiated four weeks before DENA injection and continued daily for the next 8 weeks of its injection.
- Daidzein's solvent is missing. And the solvent group is missing.
Response: Corrected. The Daidzein was dissolved in (0.1% Tween 80 in distilled water).
The authors must rewrite the animal groping and treatment periods to be understood by the reader.
Response: Corrected.
The authors didn’t clarify the sample type (serum or liver tissue) used in the oxidative stress markers.
Response: Done.
Did GGT, LDH, and glucose considered toxicity markers?
Response : We have added GGT and LDH to liver function tests in material and methods
Response: Yes, GGT (gamma-glutamyl transferase), and LDH (lactate dehydrogenase), are commonly considered toxicity markers in various research
- GGT (gamma-glutamyl transferase):
GGT is an enzyme in cell membranes, particularly the liver, kidneys, and pancreas. Elevated levels of GGT in the blood can indicate liver or biliary system dysfunction. It is often used as a marker for liver damage, cholestasis (blockage of bile flow), and certain liver diseases, such as hepatitis or cirrhosis. In some cases, increased GGT levels can also be associated with alcohol consumption [1,2].
- LDH (lactate dehydrogenase):
LDH is an enzyme that is crucial in converting lactate to pyruvate in cells. Elevated LDH levels in the blood can be a sign of tissue damage or cell death. LDH is not specific to any particular organ or tissue, so it can be a general indicator of cellular injury or damage in various parts of the body. It is commonly used as a marker for tissue damage in conditions like liver disease, heart attacks, infections, and certain cancers[1].
- In the results:
The p values of the group comparisons were missing.
Response: The P-value has been added to the results.
In the results, the authors named a group the positive control group, and this group isn’t found in the methodology section. Also, the control group is written control and is written group I. The author must uniform the group naming to not confuse the reader.
Response: The name of the group has been unified.
“GSH is a vital antioxidant and a prominent cellular reductant to assess oxidative stress in vivo”, and “Catalase is a vital antioxidant enzyme in vivo". This is not scientific writing.
Response: You are correct, and I apologize for the oversight. The statements you mentioned are not written in a scientific manner. In scientific writing, it is essential to use precise and clear language while providing appropriate context. We are sorry for the inconvenience. The sentences have been rephrased as suggested.
The results were poorly written.
Response: The results has been revised to enhance clarity and comprehension.
In lines number 228 “Lactate dehydrogenase (LDH) indicates necrosis in the living system [36].” Reference number 36 is about the detection of necrosis by the release of lactate dehydrogenase activity from cell culture media not in animal serum.
Response: Assessment of LDH is an indirect parameter to assess the extent of necrosis occurring in vivo or in situ.
In lines number 232 “However, the high dose of Daidzein was found to significantly restore the LDH activity to a value close to that of the control rats” What is meant by close to?
Response: It means that the higher dose of DZ was able to bring down the LDH activity that was comparable to the control values. However, we have rephrased the sentence for clarification.
In lines number 258 “Liver sections of DENA and low dose of Daidzein showed mild ameliorative effect” Is DENA the group I? This is very confusing and difficult to understand.
Response: Corrected in the revised version. DENA is group 2 while group 1 is the normal control group. It has been fixed in the revised version.
Fig. 10 legends need simplification.
Response: we have tried to simplify it with the current results.
- In the discussion section:
The authors write Daidzein (DZ) abbreviation in the introduction section and used that abbreviation in some places in the manuscript however referred to this abbreviation three times in the results. It must be written first, then used throughout the manuscript.
Response: Done.
The authors did not refer to Fig. 11 in the text.
Response: Done in the discussion section.
According to the authors, DZ has tremendous antioxidant and anti-inflammatory properties [41] while reference number 41 describes rat hepatocellular carcinoma induced by these chemicals. Please revise the references.
Response: Done.
In lines number 324-325, “These events cease the derogatory impacts of free radicals and control the inflammation These events cease the derogatory impacts of free radicals and control the inflammation. Hence, DZ reverses all the events triggered by the carcinogen in the target cell in the present study”, the word events were used twice and contrary to each other, which is very confusing.
“A great deal of literature suggests that DZ suggests a significant level of antitumor activity” This is a vague sentence, and only one reference [42] was added!
Response: Done
“[47] reported that these proteins induced up-regulation of two CDKIs, p27 and p21, in prostate cancer cell lines” and “Furthermore, [50] have reported that the compound triggers the upregulation” Please revise.
The discussion needs to be improved.
Response: Done
The Conclusions section is very poor. I only found two conclusive sentences, but what is reported is not consistent with what I understood from the results:
" In conclusion, the present investigation demonstrates that DZ causes apoptosis induction in the chemically induced HCC in rat models via the mitochondrial apoptotic pathway in a dose-dependent manner (Fig. 11).”, also, no figure in the conclusion section.
Response: The conclusion has been revised.
Reviewer 2 Report
The manuscript entitled "Protective Effect of Daidzein against Diethylnitrosamine/Carbon Tetrachloride-Induced Hepatocellular Carcinoma in Male Rats" by Bashandy et al. describes the preventive effects of daidzein, an isoflavone polyphenolic chemical mainly found in soybeans and other leguminous plants, against the development of hepatocellular carcinoma. The authors report several key mechanisms associated with the protective effects of daidzein, including: 1) powerful antioxidant activities, 2) anti-inflammatory capacity, and 3) improvement of lipid markers and liver test functions. The manuscript is well-written, although there are some typographical errors that need to be corrected. The results are of interest to the field; however, additional data would be beneficial to clarify and strengthen their conclusions.
11. What is the rationale for choosing only male rats? Can the authors exclude gender bias in their model?
22. The present study suggests preventive properties of daidzein in HCC development but not curative properties. To demonstrate the curative (or antitumor) activity of daidzein, I would suggest the authors first induce HCC and then initiate the treatment. If this is not possible, the authors can use a xenograft mouse model of HCC to prove the antitumor activity of daidzein.
33. In their conclusion, the authors state that daidzein induces apoptosis in chemically-induced HCC in rat models via the mitochondrial apoptotic pathway in a dose-dependent manner, without providing any proof to support their claims. I would suggest the authors perform in vitro studies using human HCC cell lines to deeply characterize the molecular mechanisms of the antitumor activity mediated by daidzein. For example, provide strong evidence of apoptosis by demonstrating caspase activation, PARP cleavage, etc.
44. Regarding abbreviation usage, the authors should pay attention to consistency throughout the text. Any abbreviated word should be defined the first time it appears in the manuscript.
55. I would also suggest the authors include a rat group that only received daidzein to demonstrate its safety. In this regard, how did the authors determine the dose of daidzein to be used in the study?
66. Based on the data shown in Figure 1 (related to body weight loss), DEN/CCI4 caused significant weight loss, indicating the toxicity of this treatment. Normally, this observation should lead to the termination of the experiment to comply with the principles of replacement, reduction, and refinement (3Rs).
The manuscript would benefit from improvements in its English language and style.
Author Response
Response to comments of Reviewer 2
Comments and Suggestions for Authors
The manuscript entitled "Protective Effect of Daidzein against Diethylnitrosamine/Carbon Tetrachloride-Induced Hepatocellular Carcinoma in Male Rats" by Bashandy et al. describes the preventive effects of daidzein, an isoflavone polyphenolic chemical mainly found in soybeans and other leguminous plants, against the development of hepatocellular carcinoma. The authors report several key mechanisms associated with the protective effects of daidzein, including: 1) powerful antioxidant activities, 2) anti-inflammatory capacity, and 3) improvement of lipid markers and liver test functions. The manuscript is well-written, although there are some typographical errors that need to be corrected. The results are of interest to the field; however, additional data would be beneficial to clarify and strengthen their conclusions.
The authors are thankful to the esteemed reviewer for valuable suggestions. We have tried to incorporate the suggestions in the revised version wherever applicable. Please find the point-to point response (green text) against each comment (red text) in the below.
- What is the rationale for choosing only male rats? Can the authors exclude gender bias in their model?
Response: The authors chose male rats because the male rats are not affected by hormonal changes that influence many in vivo analyses. However, it was our pilot study, and we shall include both genders in our upcoming study.
- The present study suggests preventive properties of daidzein in HCC development but not curative properties. To demonstrate the curative (or antitumor) activity of daidzein, I would suggest the authors first induce HCC and then initiate the treatment. If this is not possible, the authors can use a xenograft mouse model of HCC to prove the antitumor activity of daidzein.
Response: The reviewer suggestion is well-founded and could provide valuable insights into daidzein's potential curative or antitumor properties in the context of hepatocellular carcinoma (HCC). Conducting a study where HCC is induced first and then initiating daidzein treatment could help assess whether daidzein effectively reduces tumor growth or inhibits the progression of established tumors. We are planning in the future to carry out this experiment.
- In their conclusion, the authors state that daidzein induces apoptosis in chemically-induced HCC in rat models via the mitochondrial apoptotic pathway in a dose-dependent manner, without providing any proof to support their claims. I would suggest the authors perform in vitro studies using human HCC cell lines to deeply characterize the molecular mechanisms of the antitumor activity mediated by daidzein. For example, provide strong evidence of apoptosis by demonstrating caspase activation, PARP cleavage, etc.
Response: The conclusion has been replaced with a new one
- Regarding abbreviation usage, the authors should pay attention to consistency throughout the text. Any abbreviated word should be defined the first time it appears in the manuscript.
Response: Done.
- I would also suggest the authors include a rat group that only received daidzein to demonstrate its safety. In this regard, how did the authors determine the dose of daidzein to be used in the study?
Response: Determining the dose of daidzein for the study would require careful consideration of existing safety data from previous studies. Since previous studies have already declared daidzein's safety, the authors utilized that information as a starting point for their dose selection with the determination of the pharmacokinetic profile and tissue uptake of daidzein in rat serum and tissues. Lamartiniere et al. has studied toxicity in which female rats was supplied with “Diets containing 0 mg, 250 mg (low dose), and 1000 mg (high dose) daidzein/kg feed were fed to virgin female rats, starting 2 weeks before breeding and continued until the offspring were 50 days postpartum” [9]. The results indicated the high safety profile of daidzein. Furthermore another studies confirm the and safety and studied the pharmacokinetic of its metabolites [10–12], similarly, the observed results confirm the safety of daidzein. Based on the previous evidence, we decided to avoid another control group of daidzein for the sake of animal welfare and abiding with the principles of 3Rs.
- Based on the data shown in Figure 1 (related to body weight loss), DEN/CCI4 caused significant weight loss, indicating the toxicity of this treatment. Normally, this observation should lead to the termination of the experiment to comply with the principles of replacement, reduction, and refinement (3Rs).
Response: The principles of replacement, reduction, and refinement (3Rs) are ethical guidelines in animal research aimed at minimizing the use of animals, reducing their suffering, and refining experimental procedures to improve animal welfare. In this context, if the experimental model is known to cause significant weight loss, it is essential to consider the ethical implications and potential distress caused to the animals involved.
Suppose significant weight loss is observed due to the DEN/CCI4 treatment or any other intervention in an animal experiment. In that case, researchers should closely monitor the animals' welfare and consider implementing humane endpoints or terminating the experiment early to comply with the principles of the 3Rs and ensure ethical treatment of the animals involved. The usual acceptable range of humane endpoint is the loss of more than 20% per week which was not shown in the current study.
Reference:
- Pratt, D.S. Liver Chemistry and Function Tests. Sleisenger Fordtran’s Gastrointest. Liver Dis. 2 Vol. Set Pathophysiol. Diagnosis, Manag. Expert Consult Prem. Ed. - Enhanc. Online Featur. Print 2010, 1227–1237, doi:10.1016/B978-1-4160-6189-2.00073-1.
- Koenig, G.; Seneff, S. Gamma-Glutamyltransferase: A Predictive Biomarker of Cellular Antioxidant Inadequacy and Disease Risk. Dis. Markers 2015, 2015, 1–18, doi:10.1155/2015/818570.
- Elkrief, L.; Rautou, P.-E.; Sarin, S.; Valla, D.; Paradis, V.; Moreau, R. Diabetes Mellitus in Patients with Cirrhosis: Clinical Implications and Management. Liver Int. 2016, 36, 936–948, doi:10.1111/liv.13115.
- Garcia-Compean, D.; Jaquez-Quintana, J.O.; Gonzalez-Gonzalez, J.A.; Maldonado-Garza, H. Liver Cirrhosis and Diabetes: Risk Factors, Pathophysiology, Clinical Implications and Management. World J. Gastroenterol. 2009, 15, 280, doi:10.3748/wjg.15.280.
- Yao, Z.; Xu, X.; Huang, Y. Daidzin Inhibits Growth and Induces Apoptosis through the JAK2/STAT3 in Human Cervical Cancer HeLa Cells. Saudi J. Biol. Sci. 2021, 28, 7077–7081, doi:10.1016/j.sjbs.2021.08.011.
- Jin, S.; Zhang, Q.Y.; Kang, X.M.; Wang, J.X.; Zhao, W.H. Daidzein Induces MCF-7 Breast Cancer Cell Apoptosis via the Mitochondrial Pathway. Ann. Oncol. 2010, 21, 263–268, doi:10.1093/annonc/mdp499.
- Zheng, W.; Liu, T.; Sun, R.; Yang, L.; An, R.; Xue, Y. Daidzein Induces Choriocarcinoma Cell Apoptosis in a Dose-Dependent Manner via the Mitochondrial Apoptotic Pathway. Mol. Med. Rep. 2018, doi:10.3892/mmr.2018.8604.
- Kumar, V.; Chauhan, S. Daidzein Induces Intrinsic Pathway of Apoptosis along with ER α/β Ratio Alteration and ROS Production. Asian Pacific J. Cancer Prev. 2021, 22, 603–610, doi:10.31557/APJCP.2021.22.2.603.
- Lamartiniere, C.A. Daidzein: Bioavailability, Potential for Reproductive Toxicity, and Breast Cancer Chemoprevention in Female Rats. Toxicol. Sci. 2002, 65, 228–238, doi:10.1093/toxsci/65.2.228.
- QIU, F.; CHEN, X.; SONG, B.; ZHONG, D.; LIU, C. Influence of Dosage Forms on Pharmacokinetics of Daidzein and Its Main Metabolite Daidzein-7-O-Glucuronide in Rats1. Acta Pharmacol. Sin. 2005, 26, 1145–1152, doi:10.1111/j.1745-7254.2005.00187.x.
- King, R.A.; Broadbent, J.L.; Head, R.J. Absorption and Excretion of the Soy Isoflavone Genistein in Rats. J. Nutr. 1996, 126, 176–182, doi:10.1093/jn/126.1.176.
- King, R.A. Daidzein Conjugates Are More Bioavailable than Genistein Conjugates in Rats. Am. J. Clin. Nutr. 1998, 68, 1496S-1499S, doi:10.1093/ajcn/68.6.1496S.
Round 2
Reviewer 1 Report
In the revised version, the authors did not completely correct the manuscript. Writing and grammar mistakes are still there. Please find below a summary of my criticism and comments:
1. In the abstract, “the protective effect of DZ in chemically induced HCC in rat models” It must be “effect of DZ on chemically induced HCC” Not in.
· “The administration of DZ orally is initiated four weeks before HCC induction and continued until the end of the treatment period” It must be “The DZ is administered orally four weeks before HCC induction and continues during treatment.”
· “There were four treatment groups of rats (n=6) distributed as: control (group 1, without any treatment” It must be “Our study included four treatment groups: control (group 1, without any treatment.”
· “Moreover, treating HCC rats with DZ, particularly the high dose, reduced many hepatic pathological changes. All the study parameters were trending to improve with administration of DZ” It must be written “In addition, the high dose of DZ reduces many hepatological changes in HCC rats. All study parameters improved with DZ administration.”
2. The introduction section:
· Line number 42, “Cirrhosis, caused by repeated viral infections, is responsible for 75%– 80% of all cases of primary liver cancer (hepatitis B and C viruses).” What do hepatitis B and C refer to in this sentence? It must be simplified into “It is estimated that 74 to 80% of all liver cancers are caused by cirrhosis, which occurs from repeated viral infections caused by hepatitis B and C.”
· In lines number 47-48, “Aflatoxin is a strong hepatocarcinogen from Aspergillus species found in warm, humid environments in grains, maize, peanuts, or soybeans”, It must be clarified to be “Aspergillus species produce aflatoxin, a potent hepatocarcinogen, during the storage of grains, maize, peanuts, or soybeans in humid and warm environments.”
· In line number 66 “DZ is an isoflavone polyphenolic natural substance.” The author must write the full name and the abbreviation first, then use the abbreviation alone.
· The aim of the study needs grammatical improvement.
3. In the methodology section:
· The animal approval number is still missing, and the written sentence did not replace the approval assignment from the animal care committee.
· In line number 133, “At the same time, VEGF and glypican-3 (GPC3) were assayed by the same technique using kits…” What is this same technique? The authors must clarify.
· In line number 104, “Group 4: Rats treated with DENA/CCl4” did DENA/CCl4 is treatment? Please revise the methods carefully.
· Where is the reference of the DZ dose? Why did the authors choose 20 and 40 mg/kg?
· In line number 141, “The level of significance was taken at (P<0.05) for this study” Is it for this study only or all the studies?”
4. In the results:
· All the subtitles are “Effect on….” Effect of what?
· “Data showed that DENA significantly declined body weight in group 2 compared to normal rats” Declined must be replaced by reduced, decreased, or lowered.
· All the p values were written P<0.05, any statistical calculation gives specific values like 0.041, 0.002, etc… The authors must write the specific p-value for each comparison.
· In line number 147, “A significant improvement was observed in the body weight of DENA rats treated with DZ in low and high doses in group 3 and 4 (P<0.05).” compared to what group? Also, it must be in groups 3 and 4 as it is plural.
· In lines number 150-152 “An improvement in the relative body weight of the liver tissues of DENA rats (DZ treated) was observed in low and high doses compared to the control rats (P<0.05)” How could it be body weight and liver tissue?
· Many grammar errors in the manuscript were found.
· In line number 165, “Interestingly, both doses of DZ were substantially lower (P<0.05) than the HCC group” What parameter was that lowered? This is very confusing and difficult to understand.
· “Fig. 2 Major liver function markers (ALP, ALT, and AST) of the indicated treatment groups are shown as mean ± SD (n=5-6) in units/liter.” Is there a minor liver function and major liver function markers?
· “Fig. 3. Major oxidative stress markers..” as the previous comment.
· In line number 174, “Levels of Malondialdehyde (MDA) and Nitric Oxide (NO) of HCC group was significantly higher…” it must be written “Malondialdehyde (MDA) and nitric oxide (NO) levels in the hepatic tissue of the HCC group were significantly higher (P = …., and …., respectively) than those in the control group” As the authors did not mention the sample type and the sentence was poorly written, and the p values were missing in all the results.
· Please revise all the results and rewrite them as they needed extensive English editing.
· “Here, DENA rats (group 2) showed a significant decrease (P<0.05) in both the level of GSH and CAT activity of the normal control” What do you meant by of the normal control? Is it compared with or what?
· “DZ-treated group III and group IV exhibited a significant restoration (P<0.05) in GSH hepatic content in a dose-dependent manner” Compared to what?
· In line number 193, “with respect to the normal control.”??
· “DZ-treated groups III and IV significantly restored the GSH hepatic content and the hepatic CAT activity” compared to what group and restored to what?
· “Fig. 4. Major antioxidative parameters (GSH and CAT) of the indicated treatment groups” Major? And why the indicated treatment group? The figure has all the groups including the controls. Please revise.
· “All the lipid parameters (TC, TG, and LDL) showed a significantly decreasing (P<0.05) trend in the DZ-treated groups….” What do you mean by trend?
· “However, DZ was also found to elevate (P<0.05) the HDL level in the treated groups dose-dependently”, it must be “However, DZ also elevated HDL levels in the treated groups dose-dependently.”
· “Among the DZ-treated groups, LDH activity in rats of groups 3 and 4 displayed a significant (P<0.05) decline in the activity, respectively, as compared with the DENA group.” Why among the DZ? Are there any groups than 3 and 4? And respectively for what?
· “However, the high dose of DZ was found to significantly (P<0.05) ameliorated the LDH activity relative to the HCC group and to a value comparable to the control rats.” Please revise and simplify the sentence.
· In Fig. 10 legends, please add the explanation in the results text and the legend is for the main points.
5. In the discussion section:
· “The present study was aimed to see DZ's protective effect against the chemical-induced HCC in a rat animal model.” Why did the authors add “was” to “aimed”? See!! It could be evaluate or investigate? The sentence is grammatically wrong.
· “DZ treatment improved many parameters and reduced hepatic pathological changes in HCC rats by alleviating the increased inflammatory markers” many parameters! Please revise the sentence.
· “A great deal of literature suggests that DZ suggests a significant…” What do you mean by a great deal? And where are these references?
· In line number 347, “[50] reported that these proteins induced up-regulation of two CDKIs, p27, and p21, in prostate” I think the scientist’s name and year of publication must be added instead of the reference number.
· The conclusion is still poor. Please rewrite it.
· In the abstract, “the protective effect of DZ in chemically induced HCC in rat models” It must be “effect of DZ on chemically induced HCC” Not in.
· “The administration of DZ orally is initiated four weeks before HCC induction and continued until the end of the treatment period” It must be “The DZ is administered orally four weeks before HCC induction and continues during treatment.”
· “There were four treatment groups of rats (n=6) distributed as: control (group 1, without any treatment” It must be “Our study included four treatment groups: control (group 1, without any treatment.”
· “Moreover, treating HCC rats with DZ, particularly the high dose, reduced many hepatic pathological changes. All the study parameters were trending to improve with administration of DZ” It must be written “In addition, the high dose of DZ reduces many hepatological changes in HCC rats. All study parameters improved with DZ administration.”
· Line number 42, “Cirrhosis, caused by repeated viral infections, is responsible for 75%– 80% of all cases of primary liver cancer (hepatitis B and C viruses).” What do hepatitis B and C refer to in this sentence? It must be simplified into “It is estimated that 74 to 80% of all liver cancers are caused by cirrhosis, which occurs from repeated viral infections caused by hepatitis B and C.”
· In lines number 47-48, “Aflatoxin is a strong hepatocarcinogen from Aspergillus species found in warm, humid environments in grains, maize, peanuts, or soybeans”, It must be clarified to be “Aspergillus species produce aflatoxin, a potent hepatocarcinogen, during the storage of grains, maize, peanuts, or soybeans in humid and warm environments.”
· In line number 174, “Levels of Malondialdehyde (MDA) and Nitric Oxide (NO) of HCC group was significantly higher…” it must be written “Malondialdehyde (MDA) and nitric oxide (NO) levels in the hepatic tissue of the HCC group were significantly higher (P = …., and …., respectively) than those in the control group”
· And several errors were written in the comments.
Author Response
Thank you for allowing us to submit a revised draft of our manuscript entitled: “Protective effect of daidzein against diethylnitrosamine/carbon tetrachloride-induced hepatocellular carcinoma in male rats” to the Biology journal. We appreciate the time and effort dedicated to providing your valuable feedback on our manuscript. Here, we incorporate changes that reflect most of the suggestions provided by the reviewers. We have also highlighted the changes within the manuscript by tracking changes.
Here is a point-by-point response to the reviewers’ comments and concerns:
Reviewer Comments:
Reviewer 1
Comments and Suggestions for Authors
In the revised version, the authors did not completely correct the manuscript. Writing and grammar mistakes are still there. Please find below a summary of my criticism and comments:
Our appreciation goes out to whoever brought the reviewer's attention to these points.
Question 1:
- In the abstract, “the protective effect of DZ in chemically induced HCC in rat models” It must be “effect of DZ on chemically induced HCC” Not in.
Corrected as requested.
- “The administration of DZ orally is initiated four weeks before HCC induction and continued until the end of the treatment period” It must be “The DZ is administered orally four weeks before HCC induction and continues during treatment.”
Corrected as requested.
- “There were four treatment groups of rats (n=6) distributed as: control (group 1, without any treatment” It must be “Our study included four treatment groups: control (group 1, without any treatment.”
Corrected as requested.
- “Moreover, treating HCC rats with DZ, particularly the high dose, reduced many hepatic pathological changes. All the study parameters were trending to improve with administration of DZ” It must be written “In addition, the high dose of DZ reduces many hepatological changes in HCC rats. All study parameters improved with DZ administration.”
Corrected as requested.
Question 2: The introduction section:
Response 2:
- Line number 42, “Cirrhosis, caused by repeated viral infections, is responsible for 75%– 80% of all cases of primary liver cancer (hepatitis B and C viruses).” What do hepatitis B and C refer to in this sentence? It must be simplified into “It is estimated that 74 to 80% of all liver cancers are caused by cirrhosis, which occurs from repeated viral infections caused by hepatitis B and C.”
Revised as suggested.
- In lines number 47-48, “Aflatoxin is a strong hepatocarcinogen from Aspergillus species found in warm, humid environments in grains, maize, peanuts, or soybeans”, It must be clarified to be “Aspergillus species produce aflatoxin, a potent hepatocarcinogen, during the storage of grains, maize, peanuts, or soybeans in humid and warm environments.”
corrected as requested.
-In line number 66 “DZ is an isoflavone polyphenolic natural substance.” The author must write the full name and the abbreviation first, then use the abbreviation alone.
corrected as requested.
- The aim of the study needs grammatical improvement.
corrected as requested.
Question 3: In the methodology section:
Response 3:
- The animal approval number is still missing, and the written sentence did not replace the approval assignment from the animal care committee.
The correction was made as requested.
- In line number 133, “At the same time, VEGF and glypican-3 (GPC3) were assayed by the same technique using kits…” What is this same technique? The authors must clarify.
The correction was made as requested.
- In line number 104, “Group 4: Rats treated with DENA/CCl4” did DENA/CCl4 is treatment? Please revise the methods carefully.
Concerning, DENA/CCl4 was used for inducing HCC not for treatment and we apologize for this mistake.
- Where is the reference of the DZ dose? Why did the authors choose 20 and 40 mg/kg?
Thank you for your query. We apologize for any oversight in not including the reference for the DZ (Daidzein) dose selection in the manuscript. The decision to use doses of 20 mg/kg of DZ was based on previous studies [1] showing that indicated dose has antioxidant properties and is well-tolerated in experimental models. Therefore, we used this dose and also doubled it to check its dose-dependent efficacy. We have also included the reference in the revision.
.
- In line number 141, “The level of significance was taken at (P<0.05) for this study” Is it for this study only or all the studies?”
The statement "The level of significance was taken at (P<0.05) for this study" refers specifically to the significance level applied in the context of the mentioned study. It does not necessarily apply to all studies universally unless explicitly stated for each study separately. However, declaring significance when the P value is below 0.05 is conventionally acceptable. So, the level was chosen as significant in the present study.
Question 4: In the results:
Response 4:
- All the subtitles are “Effect on….” Effect of what?
The required information added.
- “Data showed that DENA significantly declined body weight in group 2 compared to normal rats” Declined must be replaced by reduced, decreased, or lowered.
The correction was made.
- All the p values were written P<0.05, any statistical calculation gives specific values like 0.041, 0.002, etc… The authors must write the specific p-value for each comparison.
We have chosen statistically standard for significance at P≤0.05. It is internationally accepted in scientific publications for reporting statistically significance.
- In line number 147, “A significant improvement was observed in the body weight of DENA rats treated with DZ in low and high doses in group 3 and 4 (P<0.05).” compared to what group? Also, it must be in groups 3 and 4 as it is plural.
The correction was made as requested.
- In lines number 150-152 “An improvement in the relative body weight of the liver tissues of DENA rats (DZ treated) was observed in low and high doses compared to the control rats (P<0.05)” How could it be body weight and liver tissue?
The sentence was rewritten as “An improvement in the liver-to-body weight ratio of DENA/CCl4 rats (Daidzein treated) was observed in low or high doses compared to the control rats (P<0.05).
Many grammar errors in the manuscript were found.
The correction was made as requested.
- In line number 165, “Interestingly, both doses of DZ were substantially lower (P<0.05) than the HCC group” What parameter was that lowered? This is very confusing and difficult to understand.
The sentence was corrected as ‘ Interestingly, both groups of Daidzein (20 or 40 mg/kg) showed substantially lower levels of ALP, ALT and AST (P<0.05) than the HCC group (Fig. 2).’
- “Fig. 2 Major liver function markers (ALP, ALT, and AST) of the indicated treatment groups are shown as mean ± SD (n=5-6) in units/liter.” Is there a minor liver function and major liver function markers?
We apologize for this mistake we have removed major.
- “Fig. 3. Major oxidative stress markers..” as the previous comment.
We apologize for this mistake we have removed major.
- In line number 174, “Levels of Malondialdehyde (MDA) and Nitric Oxide (NO) of HCC group was significantly higher…” it must be written “Malondialdehyde (MDA) and nitric oxide (NO) levels in the hepatic tissue of the HCC group were significantly higher (P = …., and …., respectively) than those in the control group” As the authors did not mention the sample type and the sentence was poorly written, and the p values were missing in all the results.
The correction was made as requested.
- Please revise all the results and rewrite them as they needed extensive English editing.
The manuscript was revised as requested.
- “Here, DENA rats (group 2) showed a significant decrease (P<0.05) in both the level of GSH and CAT activity of the normal control” What do you meant by of the normal control? Is it compared with or what?
The sentence was corrected to control group.
- “DZ-treated group III and group IV exhibited a significant restoration (P<0.05) in GSH hepatic content in a dose-dependent manner” Compared to what?
The correction was made as requested.
- In line number 193, “with respect to the normal control.”??
The sentence was corrected to control group.
- “DZ-treated groups III and IV significantly restored the GSH hepatic content and the hepatic CAT activity” compared to what group and restored to what?
The sentence was corrected to “comapred to HCC group”.
- “Fig. 4. Major antioxidative parameters (GSH and CAT) of the indicated treatment groups” Major? And why the indicated treatment group? The figure has all the groups including the controls. Please revise.
All figures has been revised as requested and the phrase “indicated treatment group” was replaced by “in rats dosed with DENA/CCl4”
- “All the lipid parameters (TC, TG, and LDL) showed a significantly decreasing (P<0.05) trend in the DZ-treated groups….” What do you mean by trend?
The sentence has been modified for clarity.
- “However, DZ was also found to elevate (P<0.05) the HDL level in the treated groups dose-dependently”, it must be “However, DZ also elevated HDL levels in the treated groups dose-dependently.”
The sentence was rewritten as requested.
- “Among the DZ-treated groups, LDH activity in rats of groups 3 and 4 displayed a significant (P<0.05) decline in the activity, respectively, as compared with the DENA group.” Why among the DZ? Are there any groups than 3 and 4? And respectively for what?
The word among was removed and the sentence was rephrased.
- “However, the high dose of DZ was found to significantly (P<0.05) ameliorated the LDH activity relative to the HCC group and to a value comparable to the control rats.” Please revise and simplify the sentence.
The sentence was paraphrased and simplified.
- In Fig. 10 legends, please add the explanation in the results text and the legend is for the main points.
The indicated sections have been edited as suggested.
Question 5: In the discussion section:
- “The present study was aimed to see DZ's protective effect against the chemical-induced HCC in a rat animal model.” Why did the authors add “was” to “aimed”? See!! It could be evaluate or investigate? The sentence is grammatically wrong.
The sentence has been rephrased.
- “DZ treatment improved many parameters and reduced hepatic pathological changes in HCC rats by alleviating the increased inflammatory markers” many parameters! Please revise the sentence.
The sentence was corrected.
- “A great deal of literature suggests that DZ suggests a significant…” What do you mean by a great deal? And where are these references?
The sentence was paraphrased and simplified.
- In line number 347, “[50] reported that these proteins induced up-regulation of two CDKIs, p27, and p21, in prostate” I think the scientist’s name and year of publication must be added instead of the reference number.
The authors and date was added to the sentence as requested.
The conclusion is still poor. Please rewrite it.
Comments on the Quality of English Language
- In the abstract, “the protective effect of DZ in chemically induced HCC in rat models” It must be “effect of DZ on chemically induced HCC” Not in.
Corrected as requested.
- “The administration of DZ orally is initiated four weeks before HCC induction and continued until the end of the treatment period” It must be “The DZ is administered orally four weeks before HCC induction and continues during treatment.”
Corrected as requested.
- “There were four treatment groups of rats (n=6) distributed as: control (group 1, without any treatment” It must be “Our study included four treatment groups: control (group 1, without any treatment.”
Corrected as requested.
- “Moreover, treating HCC rats with DZ, particularly the high dose, reduced many hepatic pathological changes. All the study parameters were trending to improve with administration of DZ” It must be written “In addition, the high dose of DZ reduces many hepatological changes in HCC rats. All study parameters improved with DZ administration.”
Corrected as requested.
- Line number 42, “Cirrhosis, caused by repeated viral infections, is responsible for 75%– 80% of all cases of primary liver cancer (hepatitis B and C viruses).” What do hepatitis B and C refer to in this sentence? It must be simplified into “It is estimated that 74 to 80% of all liver cancers are caused by cirrhosis, which occurs from repeated viral infections caused by hepatitis B and C.”
Corrected as requested.
- In lines number 47-48, “Aflatoxin is a strong hepatocarcinogen from Aspergillus species found in warm, humid environments in grains, maize, peanuts, or soybeans”, It must be clarified to be “Aspergillus species produce aflatoxin, a potent hepatocarcinogen, during the storage of grains, maize, peanuts, or soybeans in humid and warm environments.”
Corrected as requested.
- In line number 174, “Levels of Malondialdehyde (MDA) and Nitric Oxide (NO) of HCC group was significantly higher…” it must be written “Malondialdehyde (MDA) and nitric oxide (NO) levels in the hepatic tissue of the HCC group were significantly higher (P = …., and …., respectively) than those in the control group”
Corrected as requested.
- And several errors were written in the comments.
All the indicated errors have been removed in the revision.

Reviewer 2 Report
Dear Authors,
I would suggest to extensively describe the methods used in the present manuscripts and also provide the reference of the reagents (products).
Could you also correct through the manuscript that the group 2 is DENA/CCl4 and not DENA as reported elsewhere in the text
Best regards
Author Response
Response to Reviewer 2
Comments and Suggestions for Authors
Dear Authors,
Question 1: I would suggest to extensively describe the methods used in the present manuscripts and also provide the reference of the reagents (products).
Thank you to the reviewer for directing our sights to such comments.
Answer 1: The methodology has been enhanced with the required references in the revised version.
Question 2: Could you also correct through the manuscript that the group 2 is DENA/CCl4 and not DENA as reported elsewhere in the text.
Answer 2: Correction has been made.
